# HLA-DRB1 allele and autoantibody profiles in Japanese patients with inclusion body myositis

Munenori Oyama[1], Yuko Ohnuki[2], Michio Inoue[3], Akinori Uruha[4], Satoshi Yamashita[5], Sachiko Yutani[6], Jantima Tanboon[3], Jin Nakahara[1], Shingo Suzuki[7], Takashi Shiina[7], Ichizo Nishino[3], Shigeaki Suzuki[1] *

1 Department of Neurology, Keio University School of Medicine, Tokyo, Japan, 2 Department of Medial Ethics, Tokai University School of Medicine, Kanagawa, Japan, 3 Department of Neuromuscular Research, National Institute of Neuroscience, and Department of Genome Medicine Development, Medical Genome Center, National Center of Neurology and Psychiatry, Tokyo, Japan, 4 Department of Neuropathology, Charité-Universitätsmedizin, Berlin, Germany, 5 Deperment of Neurology, Graduate School of Medical Science, Kumamoto University, Kumamoto, Japan, 6 Department of Neurology, Tokai University School of Medicine, Isehara-shi, Kanagawa, Japan, 7 Department of Molecular Life Science, Basic Medical Science and Molecular Medicine, Tokai University School of Medicine, Kanagawa, Japan

* sgsuzuki@z3.keio.jp

**Data Availability Statement:** All relevant data are within the manuscript and its Supporting Information files.

## Abstract

### Introduction

Inclusion body myositis (IBM) is an idiopathic inflammatory myopathy, characterized by unique clinical features including finger flexor and quadriceps muscle weakness and a lack of any reliable treatment. The human leukocyte antigen (HLA)-DRB1 allele and autoantibody profiles in Japanese IBM patients have not been fully elucidated.

### Methods

We studied 83 Japanese IBM patients with a mean age of 69 years (49 males and 34 females) who participated in the 'Integrated Diagnosis Project for Inflammatory Myopathies' from January 2011 to September 2016. IBM was diagnosed by histological diagnosis. Various autoantibodies were screened by RNA immunoprecipitation and enzyme-linked immunosorbent assays. HLA-DRB1 genotyping was performed using polymerase chain reaction-sequence based typing. A total of 460 unrelated healthy Japanese controls were also studied.

### Results

The allele frequencies of *DRB1*01:01, *DRB1*04:10, and *DRB1*15:02 were significantly higher in the IBM group than in the healthy control group (Corrected P = 0.00078, 0.00038 and 0.0046). There was a weak association between the *DRB1*01:01 allele and severe leg muscle weakness and muscle atrophy. While hepatitis type C virus infection and autoantibodies to cytosolic 5'-nucleotidase 1A were found in 18 and 28 patients, respectively, no significant association with HLA-DRB1 alleles was observed.

**Funding:** This work was supported by JSPC KAKENHI Grant Number JP20H03592 (Shigeaki Suzuki), JSPC KAKENHI Grant Number 20K07911 (Yuko Ohnuki), and partly by Intramural Research Grant (2-5, 29-4) for Neurological and Psychiatric Disorders of NCNP (Ichizo Nishino).

**Competing interests:** Munenori Oyama, Yuko Ohnuki, Michio Inoue, Akinori Uruha, Satoshi Yamashita, Sachiko Yutani, Jantima Tanboon, Jin Nakahara, Shingo Suzuki, Takashi Shiina, and Ichizo Nishino declare no competing interests. Shigeaki Suzuki received personal fees from Alexion Pharmaceuticals, the Japan Blood Products Organization, and Asahi Kasei Medical. This does not alter our adherence to PLOS ONE policies on sharing data and materials.

## Conclusion

Japanese IBM patients had the specific HLA-DRB1 allele and autoantibody profiles.

## Introduction

Inclusion body myositis (IBM) is a slowly progressive skeletal muscle disease with unique clinical and pathological features including finger flexor and quadriceps weakness and the presence of infiltrating cytotoxic T cells in muscle [1]. Degenerative abnormalities such as numerous protein aggregates are believed to occur following the action of several autoimmunity mechanisms. In fact, IBM is categorized as an inflammatory myopathy. We created the "Integrated Diagnosis Project for Inflammatory Myopathies" to determine the correlation between autoantibodies and muscle pathology [2]. The strength of our project was to fully exclude various metabolic and genetic myopathies from inflammatory myopathies based on comprehensive histological, genetic, and chemical analyses of muscle specimen. We have demonstrated that pathological subsets of inflammatory myopathies were clearly defined by autoantibodies. Autoantibodies to cytosolic 5'-nucleotidase 1A (cN1A or 5NTC1A) were more frequently detected in IBM than in other inflammatory myopathies, even though the autoantibodies were not highly specific to it [3–7].

Certain polymorphic genes of the human major histocompatibility complex have been associated with inflammatory myopathies [8]. The strongest disease association with alleles of the human leukocyte antigen (HLA) 8.1 ancestral haplotype—*HLA-DRB1*03:01* and *HLA-B*08:01*—occurs in the clinical diagnosis of polymyositis and dermatomyositis [8]. Similarly, a disease association between *DRB1*03:01* and Caucasian patients with IBM was reported by several investigators [9–12]. However, it is noted that *DRB1*0301* is rarely detectable in the Japanese population. Although the clinical profiles of Japanese patients with IBM are similar to those of Caucasian patients [13], the immunogenetic background and autoantibody profiles have not been fully elucidated.

The purpose of the present study is to investigate autoimmune features in Japanese patients with IBM.

## Material and methods

### Patients

We studied 83 Japanese patients with IBM who participated in the 'Integrated Diagnosis Project for Inflammatory Myopathies' from January 2011 to September 2016 (S1 Table). The diagnoses were made based upon the criteria of Lloyd et al. [14]. These patients were not included in the previous reports [15,16]. We received frozen muscle biopsy blocks and blood from patients with tentative diagnoses of inflammatory myopathies, from all over Japan. Each patient's clinical information was provided by his or her referring physician, who completed detailed charts that included the clinical course and, neurological examination and laboratory findings. Previous infection with hepatitis type C virus (HCV) was determined by anti-HCV enzyme-linked immunosorbent assay (ELISA) kit (Aria HCB Ab ELISA, CTK Biotech, Poway, CA). This study was approved by the Institutional Review Boards of the Keio University (No. 20090278), National Center of Neurology and Psychiatry, and Tokai University. All of the clinical materials used in this study were obtained for diagnostic purposes with written informed consent.

## Autoantibodies detection

Frozen sera were stored at -30˚C until autoantibodies detection was performed. The screening of various autoantibodies was done by RNA immunoprecipitation and ELISAs. RNA immunoprecipitation was performed as previously described [17]. Ten-μl aliquots of serum were mixed with 2 mg of protein A-Sepharose CL-4B (Pharmacia Biotech AB) in 500 μl of immunoprecipitation buffer and incubated for 2 h. After being washed three times with immunoprecipitation buffer, antigen-bound Sepharose beads were mixed with 100 μl of HeLa cell extract ($6 \times 10^6$ cell equivalents per sample) for 2 h, and then 30 μl of 3 M sodium acetate, 30 μl of 10% sodium dodecyl sulfate, and 300 μl of phenol:chloroform:isoamyl alcohol (50:50:1, containing 0.1% 8-hydroxyquinoline) were added to extract the bound RNA. After ethanol precipitation, the RNA was resolved using a 7-M urea-8% polyacrylamide gel, and the gel was silver-stained.

ELISAs of anti-3-hydroxy-3-methylglutaryl-coenzyme A reductase (HMGCR) and anti-cN1A antibodies were performed using C-terminal recombinant HMGCR protein (Sigma) and recombinant cN1A protein (Origene, Rockville, MD) [18].

## HLA-DRB1 genotyping

Genomic DNA was extracted from peripheral blood using standard methods. HLA-DRB1 genotyping was performed using polymerase chain reaction-sequence based typing [19]. A total of 460 unrelated healthy Japanese control subjects were also studied.

## Statistical analyses

All analyses were performed using R software (version R-3.2.3) and IBM/SPSS V.20 (Armonk, New York, USA). Comparisons of relative frequencies were tested for significance using the $\chi^2$ test for 2×2 tables. Bonferroni-corrected P (corrected P) values were obtained by multiplying the observed p values by the number of DRB1 alleles (× 29). Continuous variables were compared using the Mann-Whitney U-test. Values of p<0.05 were considered significant.

## Results

### Clinical features and autoantibody profiles

Table 1 shows the clinical features and autoantibody profiles of Japanese patients with IBM (n = 83, 49 males and 34 females). The mean age of disease onset was 69 years (range 51–85). Statin exposure preceded the development of IBM in 4 patients (pitavastatin in 2 patients, atorvastatin in 1, and pravastatin in 1). Two patients had cancers within 2 years before the diagnosis of IBM (breast cancer and bladder cancer). There were five systemic autoimmune diseases including Sjögren syndrome, systemic sclerosis, rheumatoid arthritis, polymyalgia rheumatic, and microscopic polyangiitis. HCV infection was found in 18 patients (22%).

All patients exhibited objective limb muscle weakness especially in finger flexion and knee extension. Severe leg muscle weakness with grade ≤ 3/5 as assessed by manual muscle strength (Medical Research Council scale grade), usually involving the iliopsoas muscles, was seen in 37 patients. Finger flexion weakness was found in 47 patients. Dysphagia was observed in 26 patients. Although 32 patients showed neck muscle weakness, facial, cardiac, and respiratory muscle involvement was infrequent. Neurological examination revealed muscle atrophy in 67 patients. Deep tendon reflexes were decreased or absent in 40 patients. Muscle pain was reported in only 9 patients. With regard to extramuscular manifestations, skin rash, arthropathy, interstitial lung disease, Raynaud's phenomenon, and interstitial lung disease were relatively infrequent.

**Table 1. Clinical features and autoantibody profiles of 83 Japanese patients with inclusion body myositis.**

|  | Number (%) (n = 83) |
|---|---|
| Age, average | 69 years |
| Males:females | 49:34 |
| Background |  |
| Statin exposure | 4 (5%) |
| Cancer | 2 (2%) |
| Systemic autoimmune disease | 5 (6%) |
| Hepatitis C virus infection | 18 (22%) |
| Muscle symptoms |  |
| Severe leg muscle weakness | 37 (45%) |
| Finger flexion weakness | 47 (57%) |
| Facial muscle involvement | 4 (5%) |
| Dysphagia | 26 (31%) |
| Neck muscle weakness | 32 (39%) |
| Cardiac muscle involvement | 4 (5%) |
| Respiratory muscle involvement | 1 (1%) |
| Muscle atrophy | 67 (81%) |
| Myalgia | 9 (11%) |
| Extramuscular symptoms |  |
| Skin rash | 8 (10%) |
| Arthropathy | 4 (5%) |
| Raynaud's phenomenon | 2 (2%) |
| Interstitial lung disease | 2 (2%) |
| Laboratory findings |  |
| Creatine kinase, average | 676 IU/L |
| Elevated C-reactive protein | 0 |
| Antinuclear antibody positivity | 6 (7%) |
| Autoantibodies |  |
| RNA immunoprecipitation |  |
| Anti-signal recognition particle | 0 |
| Anti-aminoacyl transfer RNA synthetase | 0 |
| Anti-SSA | 3 (3%) |
| Anti-SSB | 2 (2%) |
| Anti-U1RNP | 1 (1%) |
| Anti-Ku | 1 (1%) |
| Enzyme-linked immunosorbent assays |  |
| Anti-3-hydroxy-3-methylglutaryl-coenzyme A reductase | 0 |
| Anti-cytosolic 5'-nucleotidase 1A | 28 (34%) |

The mean peak serum creatine kinase activity was 676 IU/L. Out of 83 patients, 13 had creatine kinase activity greater than 1,000 IU/L. No patients had elevation of C-reactive protein ($\geq$ 1 mg/dL). Positivity for antinuclear antibodies ($\geq$ 1:160) was detected in only 6 patients. RNA immunoprecipitation revealed no major autoantigens such as signal recognition particle (SRP) or aminoacyl transfer RNA synthetases, but some minor autoantigens were found including SSA (n = 3), SSB (n = 2), U1 RNP (n = 1), and Ku (n = 1). Additional ELISAs showed that none had anti-HMGCR antibodies, but 28 IBM patients (33%) had anti-cN1A antibodies.

## HLA association with DRB1 alleles

The DRB1 allele frequencies of the 83 Japanese patients with IBM were compared to those of 460 healthy controls (Table 2). There were significant differences between the two groups. *DRB1*01:01* (17% vs. 6%, P = 0.000027, corrected P = 0.00078), *DRB1*04:10* (8% vs. 2%, P = 0.000013, corrected P = 0.00038), and *DRB1*15:02* (24% vs. 12%, P = 0.00016, corrected P = 0.0046) were detected in the Japanese IBM patients at higher rates than in healthy controls using Bonferroni-correction. In contrast to these risk alleles, we found that *DRB1*04:06*, *DRB1*08:03*, *DRB1*09:01*, and *DRB1*12:01* were protective alleles for the development of IBM. Among these alleles, *DRB1*09:01* showed still significant after a Bonferroni correction for multiple comparisons (P = $5.4 \times 10^{-9}$, corrected P = $1.6 \times 10^{-7}$). In fact, *DRB1*09:01* was only found in one IBM patient, although it was a relatively common allele in the Japanese population.

We next examined differences in the clinical features of IBM patients between the presence and absence of the risk alleles of HLA-DRB1. IBM patients with *DRB1*01:01* (n = 28) tended

**Table 2. DRB1 allele frequencies of Japanese patients with inclusion body myositis (IBM) and healthy controls.**

| | IBM (n = 166) | Healthy controls (n = 920) | P | Odds ratio (95% confidence intervals), corrected P* |
|---|---|---|---|---|
| DRB1*01:01 | 28 (17%) | 58 (6%) | 0.000027 | 3.0 (1.7–5.0), 0.00078 |
| DRB1*03:01 | 2 (1%) | 1 (0.1%) | 0.063 | |
| DRB1*04:01 | 1 (0.6%) | 11 (1%) | 0.50 | |
| DRB1*04:03 | 3 (2%) | 23 (3%) | 0.59 | |
| DRB1*04:05 | 26 (16%) | 103 (11%) | 0.23 | |
| DRB1*04:06 | 0 | 33 (4%) | 0.0059 | 0 (0–0.64) |
| DRB1*04:07 | 0 | 5 (0.5%) | 1.0 | |
| DRB1*04:10 | 14 (8%) | 14 (2%) | 0.000013 | 5.9 (2.6–13.8), 0.00038 |
| DRB1*07:01 | 0 | 1 (0.1%) | 1.0 | |
| DRB1*08:02 | 9 (5%) | 53 (6%) | 0.86 | |
| DRB1*08:03 | 3 (2%) | 71 (8%) | 0.0038 | 0.2 (0.04–0.68) |
| DRB1*09:01 | 1 (0.6%) | 128 (14%) | $5.4 \times 10^{-9}$ | 0.04 (0.0009–0.21), $1.6 \times 10^{-7}$ |
| DRB1*10:01 | 2 (1%) | 2 (0.2%) | 0.11 | |
| DRB1*11:01 | 6 (4%) | 23 (3%) | 0.43 | |
| DRB1*12:01 | 0 | 32 (4%) | 0.0099 | 0 (0–0.66) |
| DRB1*12:02 | 2 (1%) | 19 (2%) | 0.76 | |
| DRB1*13:01 | 2 (1%) | 3 (0.3%) | 0.17 | |
| DRB1*13:02 | 6 (4%) | 68 (7%) | 0.076 | |
| DRB1*13:03 | 0 | 1 (0.1%) | 1.0 | |
| DRB1*13:07 | 0 | 1 (0.1%) | 1.0 | |
| DRB1*14:02 | 1 (0.6%) | 0 | 0.15 | |
| DRB1*14:03 | 0 | 13 (1%) | 0.24 | |
| DRB1*14:05 | 3 (2%) | 25 (3%) | 0.79 | |
| DRB1*14:06 | 4 (2%) | 12 (1%) | 0.28 | |
| DRB1*14:12 | 0 | 1 (0.1%) | 0.18 | |
| DRB1*14:54 | 1 (0.6%) | 37 (4%) | 0.027 | |
| DRB1*15:01 | 9 (5%) | 62 (7%) | 0.53 | |
| DRB1*15:02 | 40 (24%) | 114 (12%) | 0.00016 | 2.2 (1.5–3.4), 0.0046 |
| DRB1*16:02 | 4 (2%) | 6 (0.7%) | 0.052 | |

*Corrected P values are evaluated using Bonferroni-correction and are presented only in significance.

to have severe leg muscle weakness (64% vs. 35%, P = 0.01) and muscle atrophy (93% vs. 75%, P = 0.046) compared to those without *DRB1*01:01* (n = 55) (S2 Table). However, *DRB1*04:10* and *DRB1*15:02* alleles were not associated with particular clinical manifestations (S3 and S4 Tables).

## HCV infection and anti-cN1A antibodies

To disclose the relationship between the particular subsets of IBM and HLA-DRB1, 83 IBM patients were stratified by HCV infection and anti-cN1A antibodies. We compared the clinical features and DRB1 alleles between the IBM patients with and without HCV infection. IBM patients with HCV infection (n = 18) tended to have finger flexion weakness (78% vs. 51%, P = 0.0046), dysphagia (61% vs. 23%, P = 0.002), and neck muscle weakness (61% vs. 32%, P = 0.026) compared to those without HVC infection (n = 65) (S5 Table). Next, we compared the clinical features and DRB1 alleles between the IBM patients with and without anti-cN1A antibodies (n = 28 and 55, respectively), revealing no differences between the two groups (S6 Table).

Among the 83 IBM patients, 11 (13%) had both HCV infection and anti-cN1A antibodies. The frequencies of HCV infection were significantly higher in IBM patients with anti-cN1A antibodies than in those without (39% vs. 13%, P = 0.006). When we compared the clinical features and DRB1 alleles between the IBM patients with both HCV infection and anti-cN1A antibodies (n = 11) and the others (n = 72), there were no differences between the two groups (S7 Table).

## Discussion

The present study can be summarized as follows: (i) the frequencies of *DRB1*01:01*, *DRB1*04:10*, and *DRB1*15:02* were significantly higher in the Japanese patients with IBM than in healthy controls (Corrected P = 0.00078, 0.00038 and 0.0046); (ii) there was a weak association between the *DRB1*01:01* allele and clinical features (severe leg muscle weakness and muscle atrophy); and (iii) HCV infection and anti-cN1A antibodies were not associated with HLA-DRB1 alleles.

We demonstrated the strong association of *DRB1*01:01*, *DRB1*04:10*, and *DRB1*15:02* with IBM in the Japanese population. The disease's association with *DRB1*15:02* was similar to that found in a previous report [15]. Among these alleles, *DRB1*01:01* was associated with the specific clinical features of severe leg muscle weakness and muscle atrophy. However, we cannot find the significant association after the Bonferroni-correction. In this regard, a future larger study with more IBM patients would be needed to fully assess the association. Although *DRB1*03:03* was the most common allele in Caucasian patients with IBM [9–12], two studies found an additional association with IBM. Rothwell et al. indicated that *DRB1*01:01* was an independent effect associated with IBM in 252 Caucasian patients who were recruited from 11 European countries through the Myositis Genetic Consortium [9]. In addition, Rojana-Udomsart et al. showed that *DRB1*01:01* was a risk allele of IBM in 105 Australian patients [20]. We conclude from these various findings that *DRB1*01:01* allele is a common factor in the susceptibility to IBM across the populations. With regard to other autoimmune muscle disorders, the *DRB1*01:01* allele was associated with myasthenia gravis induced by D-penicillamine and with anti-melanoma differentiation-associated gene 5-positive dermatomyositis [21,22].

IBM is closely associated with autoimmunity following immune cell dysfunction in patients infected with human immunodeficiency virus or human T cell lymphotropic virus type 1 [1]. Compared to such patients, the prevalence of HCV infection was much higher in the Japanese population; it was estimated to be 3.4% in patients aged in their 60s in 2000 [23]. Our previous

study revealed that a significantly higher number of Japanese IBM patients had anti-HCV antibodies as compared with patients with polymyositis (28% vs. 5%) [16]. HCV infection is also associated with extrahepatic disorders including mixed cryoglobulinemia, Sjögren syndrome, and lymphoproliferative disorders. In our cohort, 3 of 5 IBM patients with systemic autoimmune diseases had anti-HCV antibodies. A genome-wide association study showed that *DQB1*03:03* affected the susceptibility to chronic infection with HCV in the Japanese population [24]. Since DRB1 alleles in disequilibrium with *DQB1*03:03* do not correspond to *DRB1*01:01*, *DRB1*04:10*, and *DRB1*15:02*, we think these risk alleles are associated with Japanese IBM patients independently of HCV infection.

Our comprehensive screening of autoantibodies revealed that only anti-cN1A antibodies were detected in IBM patients with a seropositive rate of 34%. The diagnostic sensitivity of anti-cN1A antibodies for IBM has varied among research groups [3–7,25]. In addition, it was also pointed out that anti-cN1A antibodies were found in other autoimmune disorders including Sjögren syndrome, systemic lupus erythematosus, and dermatomyositis [6,7]. We speculate that the discrepancy in these findings may be due to technical differences among various detection assays including ELISAs and cell-based assays. In this regard, conventional methods such as radioimmunoassay of anti-acetylcholine receptor antibodies will be required.

Our study did not detect any specific HLA-DRB1 allele associated with anti-cN1A antibodies. Similarly, Rothwell et al. reported no significant differences in HLA association between the 35 anti-cN1A-positive IBM patients and 68 anti-cN1A-negative IBM patients [9]. On the other hand, our previous study involving patients with immune-mediated necrotizing myopathy clearly showed the association between *DRB1*08:03* and anti-SRP-positive patients and that between *DRB1*11:01* and anti-HMGCR-positive patients [26]. Interestingly, *DRB1*08:03* was the risk allele for SRP-positive immune-mediated necrotizing myopathy; however, it was the protective allele for IBM. We found that anti-cN1A-postive patients tended to have HCV infection more commonly (39% vs. 13%), a finding that was the opposite of that of the report by Tawara et al. (5% vs. 27%) [25]. Further investigation will be necessary to elucidate the relationship between anti-cN1A antibodies and HCV infection.

A limitation of the present study was that we did not evaluate the outcomes of the IBM patients. IBM patients are usually unresponsive to treatment with glucocorticoids and conventional immunosuppressive agents [1]. It is possible that information of regarding HLA or other genetic risk factors will be useful for predicting a favorable response to immunotherapies or other therapies for IBM patients. In the future, an ethnographic study comparing Caucasian and Asian patients would give us important information.

## Conclusion

Japanese IBM patients had the specific HLA-DRB1 alleles and autoantibody profiles.

## Supporting information

**S1 Table. The whole dataset of 83 Japanese patients with inclusion body myositis (IBM) subjected to the present analysis.**
(DOCX)

**S2 Table. Differences in clinical features of IBM patients between the presence and absence of *DBB1*01:01*.**
(DOCX)

**S3 Table. Differences in clinical features of IBM patients between the presence and absence of *DRB1*04:10.***
(DOCX)

**S4 Table. Differences in clinical features of IBM patients between the presence and absence of *DRB1*15:02.***
(DOCX)

**S5 Table. Differences in clinical features and DRB1 alleles between the IBM patients with hepatitis C virus infection and without.**
(DOCX)

**S6 Table. Differences in clinical features and DRB1 alleles between the IBM patients with anti- cytosolic 5'-nucleotidase 1A (cN1A) antibodies and without.**
(DOCX)

**S7 Table. Differences in clinical features and DRB1 alleles between the IBM patients with both HCV infection and anti-cN1A antibodies and others.**
(DOCX)

## Author Contributions

**Conceptualization:** Shigeaki Suzuki.

**Data curation:** Michio Inoue, Jantima Tanboon.

**Formal analysis:** Munenori Oyama, Yuko Ohnuki, Sachiko Yutani.

**Funding acquisition:** Yuko Ohnuki, Ichizo Nishino, Shigeaki Suzuki.

**Investigation:** Munenori Oyama, Yuko Ohnuki, Shingo Suzuki.

**Methodology:** Akinori Uruha, Satoshi Yamashita, Takashi Shiina, Shigeaki Suzuki.

**Project administration:** Ichizo Nishino, Shigeaki Suzuki.

**Resources:** Jin Nakahara, Takashi Shiina, Ichizo Nishino, Shigeaki Suzuki.

**Software:** Yuko Ohnuki.

**Supervision:** Takashi Shiina, Ichizo Nishino.

**Validation:** Jin Nakahara.

**Visualization:** Akinori Uruha.

**Writing – original draft:** Munenori Oyama, Akinori Uruha.

**Writing – review & editing:** Shigeaki Suzuki.

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
