## [Decision Letter · Decision Letter 0]

16 Jul 2020

PONE-D-20-16597

HLA-DRB1 allele and autoantibody profiles in Japanese patients with inclusion body myositis

PLOS ONE

Dear Dr. Suzuki,

Thank you for submitting your manuscript to PLOS ONE. After careful consideration, we feel that it has merit but does not fully meet PLOS ONE’s publication criteria as it currently stands. Therefore, we invite you to submit a revised version of the manuscript that addresses the points raised during the review process.

In particular address all the critical points raised by reviewer 2 and change the statistical significance of the results and conclusions and the future plans and implications of the study accordingly.

We look forward to receiving your revised manuscript.

Kind regards,

Frederick W. Miller, MD, PhD

Academic Editor

PLOS ONE

Journal Requirements:

'Munenori Oyama, Yuko Ohnuki, Michio Inoue, Akinori Uruha, Satoshi Yamashita, Sachiko Yutani, Jantima Tanboon, Jin Nakahara, Shingo Suzuki, Takashi Shiina, and Ichizo Nishino declare no competing interests. Shigeaki Suzuki received personal fees from Alexion Pharmaceuticals, the Japan Blood Products Organization, and Asahi Kasei Medical. '

a. Please confirm that this does not alter your adherence to all PLOS ONE policies on sharing data and materials, by including the following statement: "This does not alter our adherence to  PLOS ONE policies on sharing data and materials.” (as detailed online in our guide for authors http://journals.plos.org/plosone/s/competing-interests).  If there are restrictions on sharing of data and/or materials, please state these.

Please note that we cannot proceed with consideration of your article until this information has been declared.

Reviewers' comments:

Reviewer's Responses to Questions

**Comments to the Author**

1. Is the manuscript technically sound, and do the data support the conclusions?

Reviewer #1: Yes

Reviewer #2: Yes

2. Has the statistical analysis been performed appropriately and rigorously? 

Reviewer #1: Yes

Reviewer #2: No

3. Have the authors made all data underlying the findings in their manuscript fully available?

Reviewer #1: Yes

Reviewer #2: Yes

4. Is the manuscript presented in an intelligible fashion and written in standard English?

Reviewer #1: Yes

Reviewer #2: Yes

5. Review Comments to the Author

Reviewer #1: The manuscript is clearly written and well-organized. The authors describe a study of 83 IBM patients and 460 Japanese matched-controls. High-resolution HLA DRB1 sequencing revealed multiple alleles significantly associated with IBM. Also, certain clinical signs and symptoms were likewise associated with disease-susceptibility alleles. Additional analyses of autoantibody profiles and Hepatitis C status revealed some non-significant although interesting trends. I recommend the manuscript for publication in PLOS ONE without further comment.

Reviewer #2: Many thanks for asking me to review this HLA study of patients with IBM in a Japanese poulation. The findings are of some interest and confirm an association of HLA-DRB*01:01, which has previously been confirmed in other ethnic populations in IBM. The manuscript is well written with excellent English and no obvious grammatical errors - the authors should be commended for this.

The study is somewhat small in number, and the fundamental issue with the methodology is that no attempt has been made to correct for multiple comparisons. Paragraph in table 1, there are 29 alleles therefore, if a Bonferroni correction for multiple comparisons is applied the P value must be multiplied by 29. The DRB1*01:01 and DRB1*04:10 associations do persist after this correction.

Therefore, it is more than likely that given the lack of statistical power, the described HLA/clinical associations analysed are likely by chance and again a correction for multiple comparisons must be applied here. Any associations with clinical features or serology described are at best speculative.

At the end of the study discussion, there is no research agenda or suggestions for further work based on the findings.

Surely an ethnographic study comparing Caucasian and Japanese patients would be of some interest here.

I don't feel that the Venn diagram in figure 1 is necessary and adds anything to the manuscript.

6. PLOS authors have the option to publish the peer review history of their article (what does this mean?). If published, this will include your full peer review and any attached files.

Reviewer #1: No

Reviewer #2: No

---

## [Author Response · Author response to Decision Letter 0]

24 Jul 2020

Reviewer #1

The manuscript is clearly written and well-organized. The authors describe a study of 83 IBM patients and 460 Japanese matched-controls. High-resolution HLA DRB1 sequencing revealed multiple alleles significantly associated with IBM. Also, certain clinical signs and symptoms were likewise associated with disease-susceptibility alleles. Additional analyses of autoantibody profiles and Hepatitis C status revealed some non-significant although interesting trends. I recommend the manuscript for publication in PLOS ONE without further comment.

We appreciate the kindly comment.

Reviewer #2

Many thanks for asking me to review this HLA study of patients with IBM in a Japanese population. The findings are of some interest and confirm an association of HLA-DRB*01:01, which has previously been confirmed in other ethnic populations in IBM. The manuscript is well written with excellent English and no obvious grammatical errors - the authors should be commended for this.

1. The study is somewhat small in number, and the fundamental issue with the methodology is that no attempt has been made to correct for multiple comparisons. Paragraph in table 1, there are 29 alleles therefore, if a Bonferroni correction for multiple comparisons is applied the P value must be multiplied by 29. The DRB1*01:01 and DRB1*04:10 associations do persist after this correction. Therefore, it is more than likely that given the lack of statistical power, the described HLA/clinical associations analyzed are likely by chance and again a correction for multiple comparisons must be applied here. Any associations with clinical features or serology described are at best speculative.

As suggested, we performed the statistical analyses with a Bonferroni correction for multiple comparisons. We added the following sentences.

Page 6

“Bonferroni-corrected p values were obtained by multiplying the observed p values by the number of DRB1 alleles (× 29).”

Page 9

“Further analyses using Bonferroni-corrected p value also revealed that these alleles showed a tight association with Japanese IBM patients.”

“Among these alleles, DRB1*09:01 showed still significant after a Bonferroni correction for multiple comparisons. In fact, DRB1*09:01 was only found in one IBM patient, although it was a relatively common allele in the Japanese population.” 

Page 12

“(ii) there was a weak association between the DRB1*01:01 allele and clinical features (severe leg muscle weakness and muscle atrophy);”

2. At the end of the study discussion, there is no research agenda or suggestions for further work based on the findings. Surely an ethnographic study comparing Caucasian and Japanese patients would be of some interest here.

As suggested, we modified the final sentences.

Page 14

“It is possible that information of regarding HLA or other genetic risk factors will be useful for predicting a favorable response to immunotherapies or other therapies for IBM patients. In the future, an ethnographic study comparing Caucasian and Asian patients would give us important information.”

3. I don't feel that the Venn diagram in figure 1 is necessary and adds anything to the manuscript.

We appreciate the thoughtful suggestion. We deleted the figure 1.

---

## [Editor Report · Decision Letter 1]

30 Jul 2020

PONE-D-20-16597R1

HLA-DRB1 allele and autoantibody profiles in Japanese patients with inclusion body myositis

PLOS ONE

Dear Dr. Suzuki,

Thank you for submitting your manuscript to PLOS ONE. After careful consideration, we feel that it has merit but does not fully meet PLOS ONE’s publication criteria as it currently stands. Therefore, we invite you to submit a revised version of the manuscript that addresses the points raised during the review process.

The authors claim to have adjusted P values as requested in the Methods, but have not done so in the Abstract, Results, Tables or Discussion, which are essentially unchanged.  It is possible to list the unadjusted P values and data, but the adjusted data needs to be included as the main findings, along with the implications of these findings and the minimal numbers of associations found after adjustment.  They should also state that a future larger study would be needed to more fully assess these associations.

We look forward to receiving your revised manuscript.

Kind regards,

Frederick W. Miller, MD, PhD

Academic Editor

PLOS ONE

---

## [Author Response · Author response to Decision Letter 1]

2 Aug 2020

Academic Editor

The authors claim to have adjusted P values as requested in the Methods, but have not done so in the Abstract, Results, Tables or Discussion, which are essentially unchanged. It is possible to list the unadjusted P values and data, but the adjusted data needs to be included as the main findings, along with the implications of these findings and the minimal numbers of associations found after adjustment. They should also state that a future larger study would be needed to more fully assess these associations.

We all appreciate critical suggestion. As suggested, we added and modified the following sentences. 

Page 3

“The allele frequencies of DRB1*01:01, DRB1*04:10, and DRB1*15:02 were significantly higher in the IBM group than in the healthy control group (Corrected P=0.00078, 0.00038 and 0.0046).”

“There was a weak association between the DRB1*01:01 allele and severe leg muscle weakness and muscle atrophy.”

Page 6

“Bonferroni-corrected P (corrected P) values were obtained by multiplying the observed p values by the number of DRB1 alleles (× 29).”

Page 9

“DRB1*01:01 (17% vs. 6%, P=0.000027, corrected P=0.00078), DRB1*04:10 (8% vs. 2%, P=0.000013, corrected P=0.00038), and DRB1*15:02 (24% vs. 12%, P=0.00016, corrected P=0.0046) were detected in the Japanese IBM patients at higher rates than in healthy controls using Bonferroni-correction.”

“Among these alleles, DRB1*09:01 showed still significant after a Bonferroni correction for multiple comparisons (P=5.4 × 10-9, corrected P=1.6× 10-7).”

Table 2

We added the corrected P values and foot note. 

Page 12

“(i) the frequencies of DRB1*01:01, DRB1*04:10, and DRB1*15:02 were significantly higher in the Japanese patients with IBM than in healthy controls (Corrected P=0.00078, 0.00038 and 0.0046);”

Page 13

“However, we cannot find the significant association after the Bonferroni-correction. In this regard, a future larger study with more IBM patients would be needed to fully assess the association.”

---

## [Editor Report · Decision Letter 2]

5 Aug 2020

HLA-DRB1 allele and autoantibody profiles in Japanese patients with inclusion body myositis

PONE-D-20-16597R2

Dear Dr. Suzuki,

We’re pleased to inform you that your manuscript has been judged scientifically suitable for publication and will be formally accepted for publication once it meets all outstanding technical requirements.

Kind regards,

Frederick W. Miller, MD, PhD

Academic Editor

PLOS ONE
---

## [Editor Report · Acceptance letter]

7 Aug 2020

PONE-D-20-16597R2 

HLA-DRB1 allele and autoantibody profiles in Japanese patients with inclusion body myositis 

Dear Dr. Suzuki:

I'm pleased to inform you that your manuscript has been deemed suitable for publication in PLOS ONE. Congratulations! Your manuscript is now with our production department. 

Kind regards, 

on behalf of

Dr. Frederick W. Miller 

Academic Editor

PLOS ONE